# Global structural stability and the role of cooperation in mutualistic systems

**José R. Portillo**[1,4☯]*, **Fernando Soler-Toscano**[2☯], **José A. Langa**[3,4☯]

**1** Department of Applied Mathematics I, University of Seville, Seville, Spain, **2** Department of Philosophy, Logic and Philosophy of Science, University of Seville, Seville, Spain, **3** Department of Differential Equations and Numerical Analysis, University of Seville, Seville, Spain, **4** Instituto de Matemáticas de la Universidad de Sevilla Antonio de Castro Brzezicki, Seville, Spain

☯ These authors contributed equally to this work.
* josera@us.es

**Data Availability Statement:** All real world seed-dispersal databases files are available from the web-of-life database (accession number(s). M_SD_XX). https://www.web-of-life.es/map.php?

## Abstract

Dynamical systems on graphs allow to describe multiple phenomena from different areas of Science. In particular, many complex systems in Ecology are studied by this approach. In this paper we analize the mathematical framework for the study of the structural stability of each stationary point, feasible or not, introducing a generalization for this concept, defined as *Global Structural Stability*. This approach would fit with the proper mathematical concept of structural stability, in which we find a full description of the complex dynamics on the phase space due to nonlinear dynamics. This fact can be analyzed as an informational field grounded in a global attractor whose structure can be completely characterized. These attractors are stable under perturbation and suppose the minimal structurally stable sets. We also study in detail, mathematically and computationally, the zones characterizing different levels of biodiversity in bipartite graphs describing mutualistic antagonistic systems of population dynamics. In particular, we investigate the dependence of the region of maximal biodiversity of a system on its connectivity matrix. On the other hand, as the network topology does not completely determine the robustness of the dynamics of a complex network, we study the correlation between structural stability and several graph measures. A systematic study on synthetic and biological graphs is presented, including 10 mutualistic networks of plants and seed-dispersal and 1000 random synthetic networks. We compare the role of centrality measures and modularity, concluding the importance of just cooperation strength among nodes when describing areas of maximal biodiversity. Indeed, we show that cooperation parameters are the central role for biodiversity while other measures act as secondary supporting functions.

## Introduction

Phenomena from Natural and Social Sciences are usually modeled as complex networks for which a dynamic is defined among the nodes [1–4], sometimes associated to dynamical graphs [3, 5–9], and where the study of stability is frequently a crucial fact [10, 11]. From the keynote paper from Strogatz [9], many studies have focused on possible scenarios for the long time

type=6 All synthetic computer-generated databases files are available from https://github.com/DynamicGraphSystem/StructuralStability Code is also full available.

**Funding:** This work was partially supported by FEDER Ministerio de Economía, Industria y Competitividad grant PGC2018-096540-B-I00, and Proyectos Fondo Europeo de Desarrollo Regional (FEDER) and Consejería de Economía, Conocimiento, Empresas y Universidad de la Junta de Andalucía, by Programa Operativo FEDER 2014-2020 references US-1254251 and P20-00592. The funders had no role in study design, data collection and analysis, decision to publish, or preparation of the manuscript.

**Competing interests:** The authors have declared that no competing interests exist.

dynamics of complex network with a given topology [12–15], being Population Dynamics [16–19], Economy [20–22] and Neuroscience [23–28] some of the areas where this important problem has been intensively studied. When the dynamics of the system is given by a set of differential equations, its behaviour generically depends on its global attractor [29–32], defined as information structure (IS) when its geometrical characterization is available [28, 33]. An IS includes not only the information from the topology of the graph (structural network), but other key components that are crucial to understand all possible future scenarios. Indeed, an IS is the skeleton in the phase space describing topological and geometrical structural stability in dynamical system [34]. Note that, for an autonomous system, an IS is just the detailed structure of the unique global attractor. In gradient systems, this IS induces a whole deformation of the phase space, drawing an informational landscape where the transient and asymptotic observed dynamics of the system hold [33]. This IS and informational landscape are fixed and attracting. As indicated above, they coincide with the global attractor. But, in non-autonomous systems, in which, for instance, parameters depend on time, this fixed structure and associated landscapes are also changing in time, loosing their invariance and attracting properties, but still being crucial for the description of the dynamics. This fact has been used, for instance, in Neuroscience to discriminate in detail subjects with disorders of consciousness [35]. Thus, and IS could not coincide with the standard definition of a global attractor as the object describing all the asymptotic behaviour of the system. This is why, even in an autonomous framework as we use in this paper, for attractors and IS is better if they are differentiated.

In this paper we focus on $N$-dimensional Lotka-Volterra systems used in the study of population dynamics (see, for instance, [36, 37]), but, by the Fundamental Theorem of Dynamical Systems [38], the results of this research can be extended to more general systems of differential equations. We show the dependence of dynamics on the topology of the graph, but, in addition, we claim that this fact it is only part of a more general principle: the dynamics on a graph is globally described by its associated IS, which is different from the structural base graph and whose nature is essentially informational. The IS for these systems is described as an hierarchical set of semi-stable stationary solutions linked by associated stable and unstable manifolds (see Fig 1), and informs not only on all the possible future scenarios of the system, but the way they are reached (metastability), the rate of convergence, and the zones describing phase transitions between different structures (bifurcation phenomena).

This more complex scenario leads to define a generalization of the concept of structural stability introduced in [39], in line with [40, 41], allowing for a more fine description of internal and transient dynamics in ecological systems.

A mathematical model of differential equations describes the dynamics of nodes on a mutualistic system as follows: suppose $P$ is the total number of plants and $A$ the number of animals. Plants (and animals) are in competition among them and cooperation links are set from plants to aminals and viceversa. We introduce the following system of $N = P + A$ differential equations for $S_{p_i}$ and $S_{a_i}$ describing the population density for the $i$-th species:

$$\begin{cases} \dfrac{dS_{p_i}}{dt} = S_{p_i}\left(\alpha_{p_i} - \sum_{j=1}^{P}\beta_{p_{ij}}S_{p_j} + \sum_{k=1}^{A}\gamma_{p_{ik}}S_{a_k}\right) \\[2ex] \dfrac{dS_{a_i}}{dt} = S_{a_i}\left(\alpha_{a_i} - \sum_{j=1}^{A}\beta_{a_{ij}}S_{a_j} + \sum_{k=1}^{P}\gamma_{a_{ik}}S_{p_k}\right) \\[2ex] S_{p_i}(0) = S_{p_{i0}} \\[1ex] S_{a_i}(0) = S_{a_{i0}} \end{cases} \qquad (1)$$

for each $p_i$ for $1 \le i \le P$ and $a_i$ with $1 \le i \le A$. $\alpha_{p_i}$ and $\alpha_{a_i}$ ($\alpha_i$ in short) are the intrinsic growth

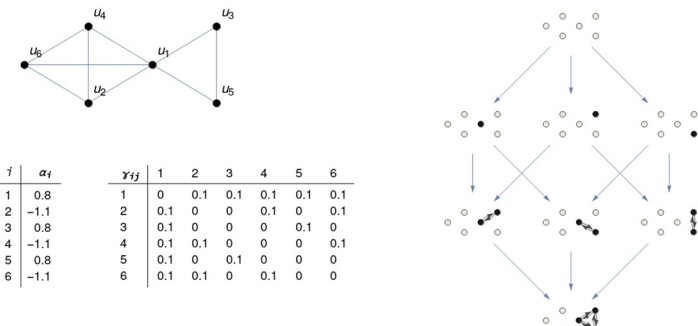

**Fig 1. Information structure.** Graph with six nodes (top left) where a dynamics is defined by means of a Lotka-Volterra cooperative system with $\alpha_i$ and $\gamma_{ij}$ parameters as shown in the tables below. The attractor associated to the system, the *information structure* (IS) is shown on the right. Its eight nodes correspond to non-negative stationary points in the dynamics of the system. These stationary points are characterised by the value of each node in the system. The nodes $u_i$ shown in white indicate that $u_i = 0$ at the corresponding point of the IS. Black nodes indicate that $u_i > 0$. Links between nodes $u_i$ and $u_j$ are those in the system (top left) where both $u_i, u_j > 0$. The blue arrows linking different points of the IS represent transitions going from one stationary solution (limit with time approaching $-\infty$) to another (when time approaches $+\infty$). For clarity, transitive arrows are not shown.

rates in the absence of competition and cooperation for plants and animals, respectively, $\beta_{p_{ij}} \geq 0, \beta_{a_{ij}} \geq 0$ denote the competitive interactions and $\gamma_{p_{ij}} \geq 0$ and $\gamma_{a_{ij}} \geq 0$ the mutualistic strengths. Structural Stability focuses on the size of the region for the intrinsic parameters $\alpha_i$ to reach optimal (maximal) biodiversity. Observe that (1) can be written as a general Lotka-Volterra model for $n$ species as:

$$\dot{u}_i = u_i \left( \alpha_i + \sum_{j=1}^{n} a_{ij} u_j \right), \qquad i = 1, \ldots, N, \tag{2}$$

or, equivalently,

$$\dot{u} = u(\alpha + Au), \tag{3}$$

with $A = (\alpha_{ij})$ the interaction (or adjacency) matrix given by

$$A = \begin{bmatrix} B^1 & \Gamma^2 \\ \Gamma^1 & B^2 \end{bmatrix}_{(P+A) \times (P+A)}. \tag{4}$$

Structural Stability for this model is introduced in [39] as a proper concept unifying the influences of network topology and parameter dependence in the system; it has been used in Theoretical Ecology to analyze robustness of biodiversity in these complex networks [42–48]. Essentially, structural stability of system (1) measures the region of intrinsic parameters of species for which we get maximal biodiversity. Note that a greater region for structural stability allows for lower values of individual intrinsic growth parameters but preserving a high level of biodiversity, pointing for robustness and resilience of species.

The study of the size of the region for intrinsic growth parameters (in our case the $\alpha_i$ parameters) for which a system reaches its optimal biodiversity (all the species present) is defined as *Structural Stability* in [39]. This is a crucial fact for the study of the robustness of biodiversity in an ecosystem, as it characterizes the borders for intrinsic growth to get maximal biodiversity.

Following the Linear Complementary Theory (LCP) associated to Lotka-Volterra systems [36, 37, 49], we introduce a partition of the phase space [50] for which we can estimate the area in which each stationary solution is globally stable, by measuring the intersection of its associated cone of biodiversity with the unit $N$-dimensional sphere [45].

However, specially in high dimensional systems, a stationary point with all its components strictly positive either does not exist, or, if this is the case, there also exists a big set of semi-stable stationary points. The presence of these stationary points is crucial for the description of the transient behaviour and metastability properties of the system, so that neglecting its study could lead to wrong conclusions. Moreover, the ways to reach a particular stationary solution are multiple, depending of the different (informational) landscapes [33] described in detail by its semistable solutions (see Fig 2).

Thus, in this paper we study the structural stability for every possible future scenario of the system. We do it in two different ways. Firstly, we consider the whole set of stationary points (asymptotically stable, semistable, or even globally unstable), and not only the globally asymptotically stable point with all components positive (see [46, 48] for a similar approach). For instance, the transition to one globally asymptotically stationary point to another by a bifurcation parameter is usually described as a sudden phenomenon, but, as we show in this paper, it is totally understandable by a careful study of the parameter region of stability for each stationary point and the way they intersect. Secondly, and maybe more important, we introduce the study of the internal dynamics for each level of biodiversity. The analytic calculation of the feasible region for any dimension has been established in Saavedra et al. [42] and in Song et al [51]. The region of maximal biodiversity is described by a cone [39, 45, 50] in the phase space for the intrinsic growth parameters $\alpha$ so that, for every $\alpha$ in this cone, the system will tend asymptotically to a stationary point with all components strictly positive. But there exists many ways to reach this global attracting state, each one defined by a different global attractor whose structure determines the transient behaviour. Indeed, in the interior of the cone of maximal biodiversity holds a rich set of different dynamical scenarios describing how species uses diverse strategies in order to reach the final stationary point. These distinct scenarios are described by different global attractors for which a topological description is available.

On the other hand, in Theoretical Ecology the study of cooperative interactions between groups of plants and pollinators / seed-dispersal / ants and how they affect to biodiversity has received an intensive research in the last fifteen years [16–19, 21]. Moreover, many studies conclude that the underlined topology of a complex network is somehow associated to the observed dynamics. Indeed, the dependence of the forwards scenarios of a system on the topology of the underlying graph is usually pointed out [13, 14, 17–19, 21, 22, 26, 52–57]. A mathematical model by a system of differential equations for mutualistic networks in Ecology was introduced in Bastolla et al. [22]. Since then, many studies have been focused on this model class, as they provide a precise analysis for a global approach to these complex phenomena. They are represented by bipartite graphs representing two kind of species (classified into two sets, plants and animals) and the cooperative links between the groups [17–19, 39, 53]. These works studied how the architecture of the network relates to biodiversity. In particular, under some conditions is observed that the more nestedness of the network, the more probability for a richer biodiversity [58]; on the other hand, it also depends on other properties of the associated graph, and cannot be considered as the only marker for a higher biodiversity [59]. Moreover, several studies suggest that this index may not play the important role in shaping the network dynamics as it was previously believed. E.g., Payrató *et al.* show that nestedness is actually an entropic consequence of the degree sequence of the mutualistic networks, and not an irreducibly macroscopic feature [60].

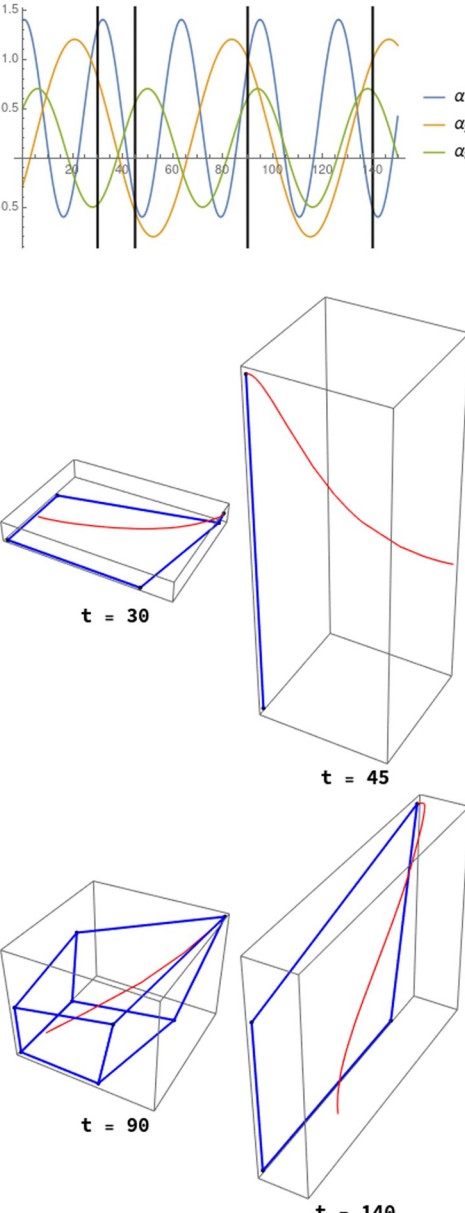

**Fig 2. The evolution of four different scenarios defined over the same graph in the same state.** A three-nodes graph is considered. A system of differential equations as (2) is defined for the three nodes. Here, $\gamma_{ji} = 0.21$ in all cases). Below, the evolution in time (red lines) from the state $(0.2, 0.2, 0.2)$ of the system, depending on the value of the $\alpha_i$ parameters which affect the behaviour of the nodes of the graph but not to its connectivity. The starting point of the red trajectories is always the same initial point but the trajectories are quite different. The changes in the trajectories are governed by the different *information structures* (figures delimited by the blue lines) in each of the dynamical systems which determine the future scenarios of the system. In each case, the trajectory goes to a special point which is the global stable solution in the phase space.

We study in detail the dependence of structural stability on several graph measures characterizing the underlying network described by adjacency matrix *A* (Results). Our findings, based on a deep computational analysis of biological and synthetic networks, conclude that cooperation parameters play the key role in biodiversity different to other graph measures such as modularity.

## Materials and methods

### Attractors as information structures

A full mathematical study of systems like (1) is developed in [49]. In particular, sufficient conditions for existence and uniqueness of solutions are provided, so defining a dynamical system $\{T(t)\}_{t\geq 0}$ for (1) which possesses a global attractor $\mathcal{A}$. The *phase space X* will be the space in which the dynamics takes place; in our case $X = \mathbb{R}^N$. We define adynamical system on $X$ as a family of non-linear operators $\{S(t)\}_{t\in\mathbb{R}^+}$,

$$S(t) : X \rightarrow X$$

$$u \in X, \quad S(t)u \in X,$$

which describes the forwards dynamics of each $u \in X$. In our case, $S(t)u_0 = u(t;u_0)$, the solution represents the solution of (1) at time $t$ with initial condition $u(0) = u_0$.

The global attractor is the central concept in dynamical system theory, since it describes all the future scenarios of the associated given phenomena. It is defined as follows [29–32, 34, 61, 62]: A set $\mathcal{A} \subseteq X$ is a global attractor for $\{S(t):t \geq 0\}$ if it is

1. compact,

2. invariant under $\{S(t): t \geq 0\}$, i.e. $S(t)\mathcal{A} = \mathcal{A}$ for all $t \geq 0$, and

3. attracts bounded subsets of $X$ under $\{S(t):t \geq 0\}$; that is, for all $B \subset X$ bounded

$$\text{dist}_H(S(t)B, \mathcal{A}) := \sup_{b\in B} \inf_{a\in\mathcal{A}} (S(t)b, a) \overset{t\to\infty}{\longrightarrow} 0.$$

Suppose $A$ in (3) belongs to *class* $S_w$ or is *Lyapunov-stable* [63], i.e., $A \in S_w$, in the sense that there exists a diagonal positive matrix $W$ such that $WA + A^T W$ is negative definite. In this case the whole structure of the global attractor for Lotka-Volterra systems can be characterized [49, 54]. Indeed, it is known that the dynamics of (3) generates an attractor, which is a structured finite set of stationary points (or equilibria) for the system, for which there exists a globally stable stationary point. The right part of Fig 1 represents the attractor corresponding to the graph on the left with the given $\alpha_i$ and $\gamma_{ij}$ parameters. Due to the informational nature of a global attractor, this attractor characterization has been defined as *information structure* (IS) in [28]. The information structure for (1) not only informs on all the stationary points of the system, but the way they are connected, showing a precise hieralchical structure by levels of information (see Fig 1 and [28, 33]).

### Global structural stability

Under the hypotheses of $A$ in (3) to be Lyapunov-stable, it is known that there exists a unique global asymptotically stable stationary point [36]. But there also exists a huge set (at most $2^N$) of actual stationary points which are determining the transient dynamics, describing the closeness to phase transitions between different scenarios of biodiversity. This information is contained in the IS described above.

### Convex cone partition of $\mathbb{R}^N$

Let us introduce the precise definitions related to global structural stability: let $D = \{1, \ldots, n\}$, $I \subset D$ and $J = D \backslash I$. Let $A$ a Lypaunov-stable matrix and $B_{\cdot j} = -A_{\cdot j}$ for $j \in J$ and $B_{\cdot j} = -I_{\cdot j}$ (negative

of identity matrix) for $j \in I$, where $B_{\cdot j}$ the $j$-column of matrix $B$. Define the *convex cone* as

$$pos(B_{\cdot 1}, \ldots, B_{\cdot N}) = \{\alpha \in \mathbf{R}^N : \alpha = r_1 B_{\cdot 1} + \ldots + r_n B_{\cdot N}; \ r_i \geq 0\}. \tag{5}$$

Then, for any $\alpha \in pos(B_{\cdot 1}, \ldots, B_{\cdot N})$, the unique globally stable stationary solution of (3) $u_J^* = \{u^*(1), \ldots, u^*(N)\}$ satisfies [36]:

$$\begin{aligned} \{j \in D/ \ u^*(j) > 0\} &= J, \text{ and} \\ \{i \in D/ \ u^*(i) = 0\} &= I. \end{aligned}$$

This is an important result as, given any possible stationary point of the system, there exists an associated convex cone as described in $pos(B_{\cdot 1}, \ldots, B_{\cdot N})$ such that, when $\alpha \in pos(B_{\cdot 1}, \ldots, B_{\cdot N})$, this stationary point is globally asymptotically stable [36].

For instance, if we consider a 4D Lotka-Volterra system, and $J = \{1, 2\}$ (so that $I = \{3, 4\}$,) the portion (i.e., the cone $C_J$) of the $\mathbf{R}^4$ space for parameter $\alpha$ assuring that the global asymptotic stationary point is of the form $u_J^*$ is given by

$$C_J = \{\alpha \in \mathbf{R}^4 : \alpha = r_1(-A_{\cdot 1}) + r_2(-A_{\cdot 2}) + r_3(-I_{\cdot 3}) + r_4(-I_{\cdot 4}), \ r_i > 0\}.$$

Even more interesting, the sixteen possible cones ($2^4$) $C_J$, for all possible $J$– choices, form a partition of $\mathbf{R}^4$, i.e., the union of cones fills all the space and there is no intersection between their interiors.

Fig 3 shows an example of the calculation for a graph of five nodes ($n_1$, $n_2$ at the left and $n_3$, $n_4$, $n_5$ at the right). Competition (dashed lines) is assumed between every pair of nodes on the same side of the graph and cooperation (solid lines) exists when there is a link $n_i \leftrightarrow n_j$ joining nodes of different sides. Note that when considering cooperation relationships is a bipartite

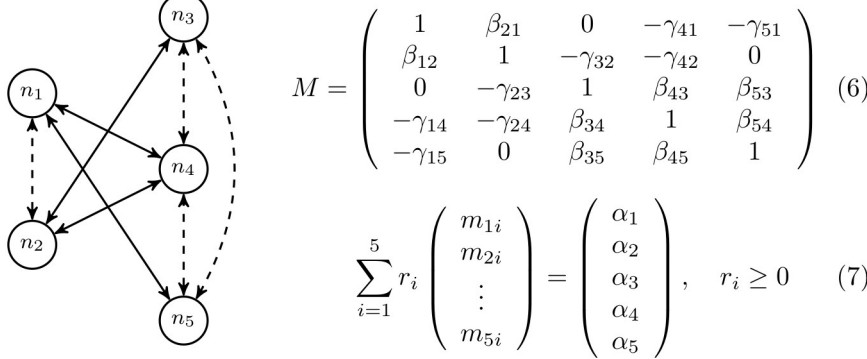

$$M = \begin{pmatrix} 1 & \beta_{21} & 0 & -\gamma_{41} & -\gamma_{51} \\ \beta_{12} & 1 & -\gamma_{32} & -\gamma_{42} & 0 \\ 0 & -\gamma_{23} & 1 & \beta_{43} & \beta_{53} \\ -\gamma_{14} & -\gamma_{24} & \beta_{34} & 1 & \beta_{54} \\ -\gamma_{15} & 0 & \beta_{35} & \beta_{45} & 1 \end{pmatrix} \tag{6}$$

$$\sum_{i=1}^{5} r_i \begin{pmatrix} m_{1i} \\ m_{2i} \\ \vdots \\ m_{5i} \end{pmatrix} = \begin{pmatrix} \alpha_1 \\ \alpha_2 \\ \alpha_3 \\ \alpha_4 \\ \alpha_5 \end{pmatrix}, \quad r_i \geq 0 \tag{7}$$

**Fig 3. The cone of maximal biodiversity and structural stability.** Left: Example (2 + 3)-bipartite graph with two nodes on the left and three nodes on the right. Competition (dashed lines) is between all elements on the same side, while cooperation occurs between elements on different sides for which there is an edge represented by a solid line. Cooperative relationships form a bipartite graph with two groups of nodes (left and right). Top right: Connectivity matrix $-M$ of the system. Competition parameters $\beta_{ij} > 0$ are set for all pairs of nodes $n_i$, $n_j$ in the same group (both at the left or the right). Cooperation parameters $\gamma_{ij} > 0$ exist for nodes $n_i$, $n_j$ in different groups (one in the left and the other in the right) only when there is an arrow $n_i \leftrightarrow n_j$ in the graph. In our experiments $\beta_{ij} = \beta_{ji}$ and $\gamma_{ij} = \gamma_{ji}$. Bottom right: Equation to determine if a given point $\bar{\alpha} = (\alpha_1, \ldots, \alpha_5) \in \mathbf{R}^5$ is in the maximal biodiversity cone. If there exist some $r_i \geq 0$, $1 \leq i \leq 5$, which verify (7), then $\bar{\alpha}$ is in the *maximal biodiversity cone* of $M$. To ensure the existence of a solution, the sum of the absolute values of each row or column of $M$ (including the 1 in the diagonal) must be always lower than 2. This is equivalent to bound to 1 the sum of weights of all edges adjacent to each node of the graph (node degree bounded to 1). Given $M$, *structural stability* is defined as the proportion of points $\bar{\alpha} \in \mathbf{R}^5$ for which (7), has a solution. Since the cones are centred at 0, considered points can be limited to those on the surface of the $\mathbf{R}^5$ sphere of radius 1 centred at 0.

graph. In general, $\beta_{ij}$ is not necessarily equal to $\beta_{ji}$ and the same for $\gamma_{ij}$ and $\gamma_{ji}$. Matrix $M$ in (6) (note that, in our case $M = -A$) contains all connectivity parameters. Observe that the diagonal is 1 and 0 represents no interaction between nodes. Given $M$, a certain $\bar{\alpha} \in \mathbf{R}^N$ is in the maximal biodiversity cone when there exist $r_1, r_2, \ldots, r_N \geq 0$ verifying (7). Structural stability for $M$ is defined as the proportion of $\bar{\alpha} \in \mathbf{R}^N$ in the maximal biodiversity cone; i.e., the structural stability of $M$ is equal to the proportion of points of the $\mathbf{R}^N$ sphere of radius 1 centred at 0 in the maximal biodiversity cone.

## Results

### Global structural stability

In this section, we study the structural stability of each stationary point in the system, independently of their stability properties. This is referred as *Global Structural Stability*. For (3), the zero solution is globally unstable, each stationary $u_j^*$ point belongs to an informational level $E_i$ and possesses stable and unstable directions, and there exists just one stationary feasible point $u^*$ in the lower level which is globally stable (see Fig 1).

To illustrate the description of global structural stability, consider a two-dimensional cooperative system given by

$$\begin{cases} \dot{u}_1 = u_1(\alpha_1 - u_1 + au_2) \\ \dot{u}_2 = u_2(\alpha_2 - u_2 + bu_1) \end{cases} \tag{8}$$

with $\alpha_i \in \mathbf{R}$ and $a, b > 0$. For a fixed network of connections in the system (given by values of the $a$ and $b$ parameters), the intrinsic growth rate of each species plays a crucial role. Indeed, a convex cone of $\alpha$ parameters in (8) is associated to these stationary points, and all of these convex cones form a partition of $\mathbf{R}^2$ [50], i.e., each cone has a nonvoid interior, the union of all cones is $\mathbf{R}^2$ and each pair of the interior of cones is disjoint (see Fig 4).

This means that a given vector $\alpha$ of (8) belongs either to just the interior of one cone (determining the feasible stationary point, and so the future biodiversity of the system) or to the intersection of cones, made by rich manifolds showing a phase transition and a high sensibility to bifurcation scenarios in biodiversity.

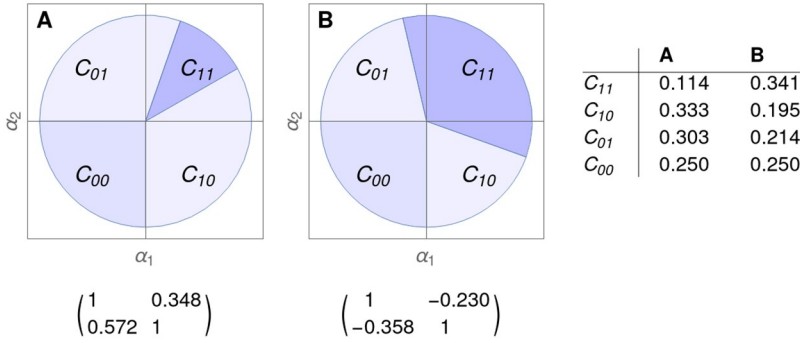

**Fig 4. Description of cones for *alpha* parameters associated to 8 with 2 × 2 matrices as indicated. A.** Competitive case. **B.** Cooperative case. Note that there are four regions $C_{ij}$, $i, j = 0, 1$, each for the four possible stationary points. If $\alpha \in C_{ij}$, the globally asymptotically stable point $u^*$ for (8) has the positive components pointed by $ij$, i.e., $\alpha \in C_{11}$ means that in $u_1^*, u_2^* > 0$. Note that borders of each cone are bifurcation lines, in the sense that a sudden attractor bifurcation occurs when passing through this border. Moreover, inside each cone, there exist interesting zones marking different attractor structures with the same globally asymptotically stable solution, which is also crucial for the study of the structural stability.

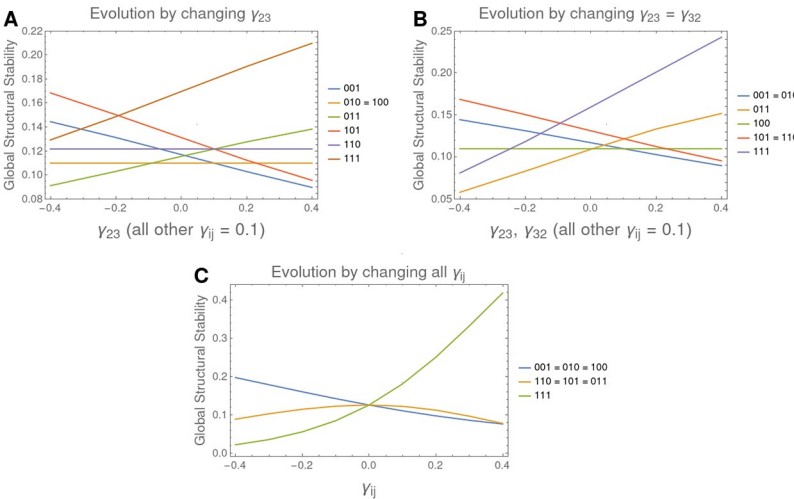

**Fig 5. Evolution of global structural stability on parameters on a 3D LV system.** A. We observe the evolution of the size of the cones when changing parameter $\gamma_{23}$ from competition (0.4) to cooperation values (−0.4). We observe that cones of maximal biodiversity ($u_{111}$) and the cone associated to stationary point $u_{011}$ behaves monotonically increasing with $\gamma_{23}$. B. The same result, now simultaneously changing $\gamma_{23}$ and $\gamma_{32}$. Note that the cones for $u_{011}$ and $u_{111}$ now grow faster, while other cones with constant size now decreases (as those associated to $u_{010}$ and $u_{110}$). C It is shown the Global Structural Stability from a competitive system to a cooperative one, by changing all the parameters in matrix $A$. Note that, among all, it is the cone with maximal biodiversity the only one increasing with an income of cooperation in the system.

The global structural stability of a system allows to study bifurcation and transitions between different biodiversity scenarios. Indeed, the borders, which are now mathematically well defined, of each cone are critical zones for the sudden transition to one biodiversity scenario to a different one. Moreover, we can also observe in a global way the dependence of the cone partition on the parameters of the system (see Fig 5).

In Fig 6 we show the different cones for a three dimensional Lotka-Volterra system. Note, once more, that the union of cones forms a partition of $\mathbf{R}^3$.

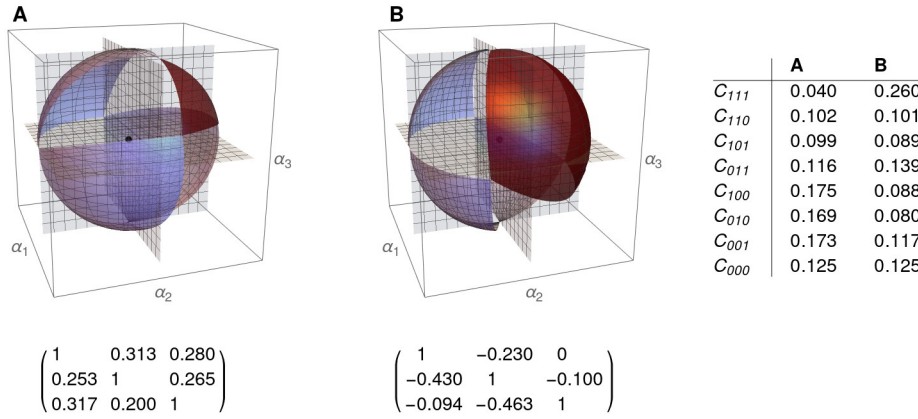

**Fig 6. Two representations of the eight cones describing global structural stability. A.** a 3D competitive LV system, with the cone of maximal biodiversity (dark blue). **B.** a 3D cooperative LV system, with a bigger cone of maximal biodiversity, pointing out the key role of cooperation in biodiversity.

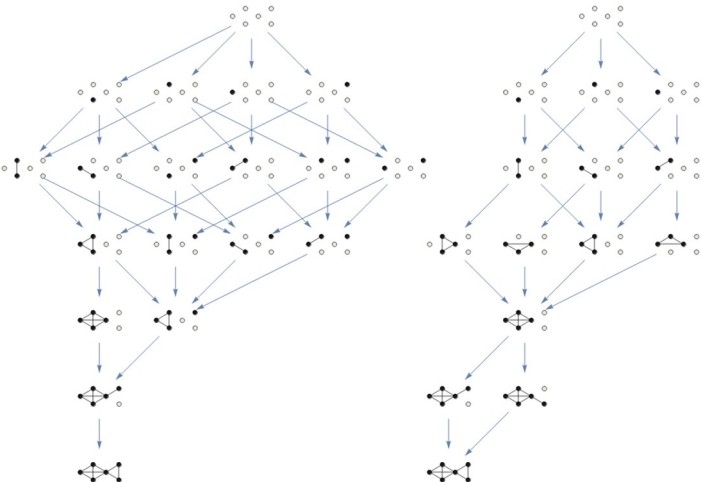

**Fig 7. Two different information structures for a six dimensional cooperative LV system, associated to the same cooperative values between nodes (same adjacency matrix) and with different values of the intrinsic parameter $\alpha_i$.** Note that both structures have the same kind of globally asymptotically stationary point, the one with the six components strictly positive. However, these structures produce a very different phase space deformation (information field) in both cases. Indeed, all the transient behaviour of every solution will follow the metastability imposed by the structure. These complex graphs are just two of the many that coexist within the cone of maximal biodiversity that describes the structural stability in the sense of [39] for this system.

We will refer to *Global Structural Stability* as the study of the properties of regions describing the different possible scenarios of biodiversity in an ecological system. The size of the cone-partition of the phase space and its dependence on the structural network of connection between species suppose core subjects in this analysis.

The study of the structural stability for all possible asymptotic realization informs, in a holistic way, on the robustness of a particular future scenario and the way it is reached.

**Topological structural stability in the interior of cones.** In Fig 7 we can can see two of the different possibilities converging to a six-node asymptotically stable stationary point. These structures would belong to the same cone of biodiversity, but the different characterization infers important qualitative dynamics differences. Each of these structures defines a region in the interior of this cone, making the structure to be stable under perturbation [34], causing that the union of regions, the whole cone, is also structural stable. Indeed, the structural stability of the cone of maximal biodiversity can be analyzed as the union of structural stability regions inside the cone describing different transient scenarios. It is the structural stability of these complex graphs which is dynamically relevant, in the sense to cause the dynamics, very different from region to region even all belonging to the cone of maximal biodiversity.

In Fig 8 we observe the region of maximal biodiversity for a three-node cooperative LV system. This region is subdivided into different areas, each one corresponding to a different information structure. These are the minimals regions which are stable under perturbation in the sense of [34, 40], as they characterize the phase space and its transitions to other topological configurations of the space in the interior of the cone of maximal biodiversity. Note that this richness of possibilities is unobservable in the competitive case, showing (once more) the role of cooperation in the generation of different scenarios.

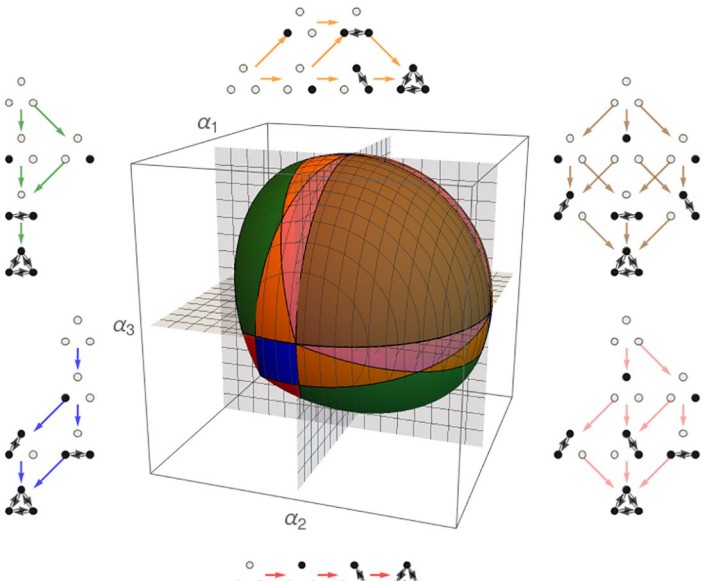

**Fig 8. Cone of maximal biodiversity for a three-nodes LV cooperative system.** Each color corresponds to a kind of information structure, also represented. The union of all regions produces the cone where structural stability is defined in the sense of [39]. Observe the rich presence of different attractors in the same cone, producing different dynamics with a same globally stable point. Each of these attractors is structurally stable under autonomous and nonautonomous perturbations [34], producing the structural stability of the whole cone. These coloured areas are the minimal structural stable zones, so defining with full precision the observed dynamics, in particular metastability effects, and predicting sudden bifurcation in structures.

## Graph characterization and structural stability

Antagonism/competition ($\beta_{ij}$ parameters) is established etween all pairs of nodes in the same set and mutualism/cooperation ($\gamma_{ij}$) between nodes of different sets. Competition relations are constant in most experiments (random in some cases), only the cooperation part changing across all graphs. If we only look at the cooperation relationships, the graphs are bipartite. We will call these graphs as $\gamma$-bipartite and they have a similar structure to the example in Fig 3 where the weights of edges connecting different sides correspond to cooperation $\gamma_{ij}$ parameters and inside each group of nodes (left and right) competition $\beta_{ij}$ parameters are fixed. This structure is familiar in the study of mutualistic ecological networks [18, 19, 64], like plants-pollinators, seed-dispersal, or plants-ants relationships. As explained above, $\beta_{ij} = \beta_{ji}$ and $\gamma_{ij} = \gamma_{ji}$. Also, $0 < \beta_{ij}, \gamma_{ij} < 1$ and $\Sigma_j(\beta_{ij} + \gamma_{ij}) < 1$ for each $i \in \{1, \ldots, n\}$.

Given a connectivity matrix $M$ (like the example in Fig 3) we have estimated structural stability by using thesoftware by Jacopo Grilli presented in [43].

We have explored the relation between structural stability and several graph measures. We focus on connected graphs representing networks with nodes divided into two sets. Experiments use networks of size 10 + 10 and 20 + 30. Note that the adjacency matrix of our graphs is matrix $A$ in (3). Two kinds of experiments on biological and synthetic networks (Table 1) are performed in two sizes (20 and 50 nodes). To study the dependence of structural stability on several graph measures, Spearman correlation coefficient $\rho$ is used (Table 2). It it a non-parametric measure on the correlation between two (discrete or continuous) random variables [65]. To compute $\rho$, data are ordered and replaced by their respective order.

Edges are weighted in different ways. In the first set of experiments, real mutualistic networks of seed-dispersers obtained from the *Web of Life* (http://www.web-of-life.es/) [66] have

**Table 1. Summary of the experiments with graphs of 10 + 10 (`M_SD_20`) and 20 + 30/30 + 20 (`M_SD_50`) nodes extracted from seed-dispersal biological networks [66], and synthetic graphs of 10 + 10 (`G`*) and 20 + 30 nodes (`E`*), with different connectivity values. Structural stability is approximated with an error margin of 1% using the software described in [43].**

| Experiment | Antagonism coefficients | Cooperation coefficents |
| --- | --- | --- |
| `M_SD_20` | 0.01 | normalized from real network |
| GB | 0.01 | $\gamma_{ij} = (i + j) * k$ (normalized) |
| GC | random | random |
| GD | 0.01 | random |
| `M_SD_50` | 0.002 | normalized from real network |
| EB | 0.002 | $\gamma_{ij} = (i + j) * k$ (normalized) |
| EC | random | random |
| ED | 0.002 | random |

been used to get 10 $\gamma$-bipartite connected graphs of 10 + 10 nodes in the experiment `M_SD_20` and 4 $\gamma$-bipartite connected graphs of 20 + 30 (or 30 + 20) nodes in the experiment `M_SD_50`. In the second set, 500 $\gamma$-bipartite connected graphs of 10 + 10 nodes have been generated for experiments GB, GC and GD, and 500 $\gamma$-bipartite connected graphs of 20 + 30 nodes for experiments EB, EC and ED.

Experiments `M_SD_20` and `M_SD_50` have been performed on seed-dispersal networks from the *Web of Life* [66]. This dataset contains 16 weighted databases of mutualistic seed-dispersal with a number of species varying from 23 and 121 (from 4 to 50 plants and from 8 to 88 animals). Given that the network size may influence on the associated graph parameters, we have selected those networks with more than 10 plants and 10 animals for `M_SD_20` and more than 20 plants and 30 animals (or vice-versa) for `M_SD_50`. From each network in these two sets, we choose a subnetwork of the desired order (resp. 10 + 10 and 20 + 30). To build these networks of the same size, we selected the plants and animals with more observed interactions.

Graphs associated with experiments `M_SD_20` and `M_SD_50` have been built by setting the mutualistic values $\gamma_{ij}$ to 0.75 times the number of observations divided by the sum of observations of each specie. Antagonistic $\beta_{ij}$ values are all set to 0.01 in `M_SD_20` and 0.002 in `M_SD_50`. A summary of these values can be found on Table 1.

**Table 2. Spearman correlation coefficients between structural stability and several graph measures for the experiments with *biological* networks, considering 10 $\gamma$-bipartite graphs of 10 + 10 nodes given from selected subnetworks of seed-dispersal networks (first row) and 4 $\gamma$-bipartite graphs of 20 + 30 or 30 + 20 nodes given from selected subnetworks of seed-dispersal networks (fifth row). The rest of rows correspond to the several sets of synthetic random graphs: 500 $\gamma$-bipartite graphs of 10 + 10 nodes (rows 2-4) and 500 $\gamma$-bipartite graphs of 20 + 30 nodes (rows 6-8). See Table 1 for details on how $\gamma_{ij}$ and $\beta_{ij}$ are set.**

| Experiment | Degree centrality | | | Optimal modularity |
| --- | --- | --- | --- | --- |
| | (mean) | (mutualism) | (antagonism) | |
| *`M_SD_20`* | *1.000000* | *1.000000* | — | *-0.042424* |
| GB | 0.994869 | 0.994869 | — | -0.580607 |
| GC | 0.999332 | 0.999613 | 0.0353016 | -0.570839 |
| GD | 0.999670 | 0.999670 | — | -0.6066086 |
| *`M_SD_50`* | *1.00000* | *1.00000* | — | *-0.600000* |
| EB | 0.999424 | 0.999424 | — | -0.625978 |
| EC | 0.997284 | 0.998848 | 0.051100 | -0.622205 |
| ED | 0.999399 | 0.999399 | — | -0.609350 |

Studies `GB`, `GC` and `GD` (G* in short) have been carried out on synthetically generated networks. In all these G* studies, each network is built by creating a random bipartite graph of 10 + 10 nodes with a number of edges drawn from a normal distribution whose mean (52.6) and standard deviation (17.2) correspond to that of the population of edges in the graphs associated with the networks in the `M_SD_20` study.

Similarly, `EB`, `EC` and `ED` studies (E* in short), are based on synthetically generated networks which are obtained by creating random bipartite graphs of 20 + 30 nodes with a number of edges obtained from a normal distribution whose mean (157.25) and standard deviation (76.8) correspond to that of the population of edges in the graphs associated with the seed-dispersal networks of the `M_SD_50` study.

Study `GB` is done by weighting graph edges as $\gamma_{ij} = (i + j)^* k$ and then normalising by dividing the matrix by the maximum sum of rows and columns and multiplying all resulting matrices by a suitable constant so that the mean of the degrees is the same as the average $\gamma_{ij}$ in experiment `M_SD_20`. The goal is to obtain a collection of networks whose average degree is uniformly increasing, avoiding the random effect of the construction of the bipartite graphs in G*. The value of $\beta_{ij}$ is 0.01 as in the graphs of the experiment `M_SD_20`.

Studies `GC` and `GD` set $\gamma_{ij}$ randomly, using a normal distribution whose mean and standard deviation are equal to that of the experiment `M_SD_20`. When a negative random value is produced, $\gamma_{ij} = 0.002$. In the `GD` experiment, the values of $\beta_{ij}$ are all equal to 0.01 (like in `M_SD_20`). In `GC`, $\beta_{ij}$ is calculated from a probability distribution with mean 0.01 and standard deviation 0.002. When a negative random value is produced, $\beta_{ij} = 0.002$.

In a similar way, the study `EB` is done by weighting graph edges as $\gamma_{ij} = (i + j)^* k$, then normalising by dividing by the maximum sum of weights of rows and columns and multiplying all the adjacency matrices of the graphs by an appropriate constant so that the mean of the degrees is the same as the mean of the $\gamma_{ij}$ of the graphs in the experiment `M_SD_50`. In the other hand, studies `EC` and `ED` set $\gamma_{ij}$ randomly, using a normal distribution whose mean and standard deviation are equal to those of experiment `M_SD_20`, with $\beta_{ij}$ fixed and equal to 0.002 in the experiment `ED` and getting $\beta_{ij}$ values from a probability distribution with mean 0.006 and standard deviation 0.001. When negative values are randomly produced, $\beta_{ij} = 0.001$.

These studies aim at a comprehensive exploration of the relationship between structural stability and several network measures. This is why the construction of graph sets is intended to cover as much of the search space of connected graphs as possible.

In all these graph sets, we have studied the correlation between structural stability and several graph measures. Our aim is to get insight into the criteria producing connectivity matrices with higher structural stability. Degree centrality and optimal modularity are studied in all experiments.

Table 2 shows the Spearman correlation coefficient between these parameters and structural stability, estimated by using the software in Grilli [43] (https://github.com/jacopogrilli/feasibility) with an error margin of ±1%.

**Centrality measures.** Centrality measures determine the relative importance of a node within a network [67, 68]. Being able to recognise the centrality of a node can help to determine, for example, the impact of a person involved in a social network, the relevance of a room in a building represented in spatial syntax, the importance of a road in an urban network, or the essential components of a computer network.

Centrality is not an intrinsic attribute of the nodes or actors in a network, such as temperature, monetary income, etc., but a structural attribute, that is, an assigned value which strictly depends on its location in the network. Centrality measures the contribution of a node according to its location in the network, regardless of whether its importance, influence, relevance or prominence is being assessed.

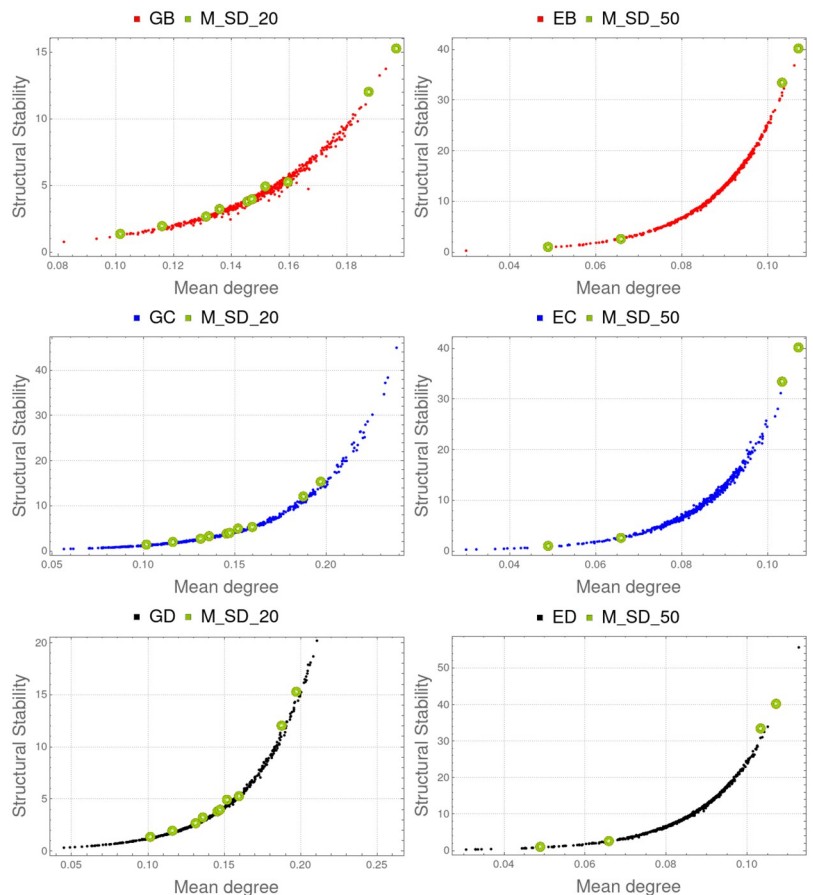

**Fig 9. Degree centrality and structural stability.** Left (top to bottom): Comparison of `M_SD_20` with `GB`, `GC` and `GD`. Right (top to bottom): Comparison of `M_SD_50` with `EB`, `EC` and `ED`. Spearman correlation coefficient is greater than 0.994 in all these cases (Table 2). In these graphs, degree centrality grows with the sum of the cooperation ($\gamma_{ij}$) and competition ($\beta_{ij}$) parameters.

*Degree centrality* corresponds to the number of links that a node has with the others [69]. It provides the clearest relationship with structural stability in most cases. As degree centrality is a measure of the nodes, we use for each graph the mean degree (equivalent to the sum) of its nodes.

Correlation between mean degree and structural stability is very strong in all experiments (Table 2). See Fig 9 for a graphical representation of these results.

These results suggest that structural stability (ability to produce maximal biological diversity) is related to degree centrality or, equivalently, to the sum of node degrees. This sum includes both mutualism and antagonism coefficients. However, we can observe that the structural stability grows as the sum of the cooperation parameters $\gamma_{ij}$ grows and has no apparent relationship with the sum of the antagonism parameters $\beta_{ij}$ (see Table 2 and Fig 10). Because of this, instead of considering the mean (or, equivalently, the sum of all antagonistic and mutualistic coefficients) we can conjecture that structural stability is mainly related to $\Sigma\gamma_{ij}$, the cooperation within network nodes. Since the antagonistic coefficients are lower in absolute value than the mutualistic ones, it is possible that their effect on structural stability can only be observed when they have a similar weight. Further experiments may confirm or refute this hypothesis.

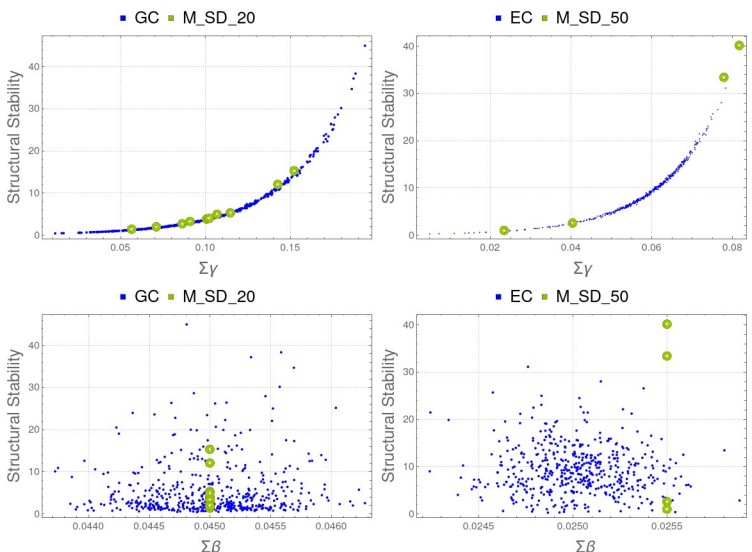

**Fig 10. Sum of cooperation $\Sigma\gamma_{ij}$ (top row) and competition $\Sigma\beta_{ij}$ (bottom) versus structural stability in graphs with random antagonistic edge weight: GC (left) and EC (right).** Correlations for $\Sigma\gamma_{ij}$ and $\Sigma\beta_{ij}$ are shown in Table 2. Results indicate that the relationship of structural stability is direct to the sum of the cooperation coefficients but it is not correlated with the sum of the competition ones.

To test our conjecture, we have calculated the Spearman correlation between structural stability and $\Sigma\gamma_{ij}$ alone (and the same for $\Sigma\beta_{ij}$) for the experiments with random values of $\beta_{ij}$ and $\gamma_{ij}$, i.e., GC and EC studies. Results are shown in Table 2 and Fig 10, confirming that structural stability (maximal biological diversity) is in direct relation to the sum of the cooperation coefficients. This is a relevant result because it shows that the more cooperation between network nodes, the greater structural stability is produced. A similar study has been done in [43], showing also that cooperation is a core process for feasibility. Other measures discussed below focus on how the distribution of cooperation affects structural stability.

**Modularity.** *Modularity* measures the division of a network into communities (groups, clusters or modules) [70]. Its value (between −1/2 and 1) is relative to a partition of the graph nodes into different communities. It is defined as the fraction of the edges within the communities minus the expected fraction if edges were set at random. The maximum value for a graph (considering all possible partitions) is called *optimal modularity*. Networks with high optimal modularity (close to 1) have high connectivity between nodes within the same community and low connectivity between different communities. Modularity is 1 just in case that communities are complete graphs with no links between them. Some biological networks have high degrees of optimal modularity [25]. Graphs with a very specific structure (grid or similar) have negative optimal modularity.

We have used the C library iGraph [71] which implements the Newman [70] algorithm to compute optimal modularity. As Table 2 show, Spearman correlation is relatively strong (up to 0.57) in all experiments except M_SD_20, in which is very low (see Table 2 and Fig 11).

## Discussion

In general, to study the long-time behaviour of a dynamical system, it is a common way to analyse its stationary points and associated stability. An IS contains this information as a whole, all the possible scenarios at the same time, their stability, their robustness, and, crucially, the

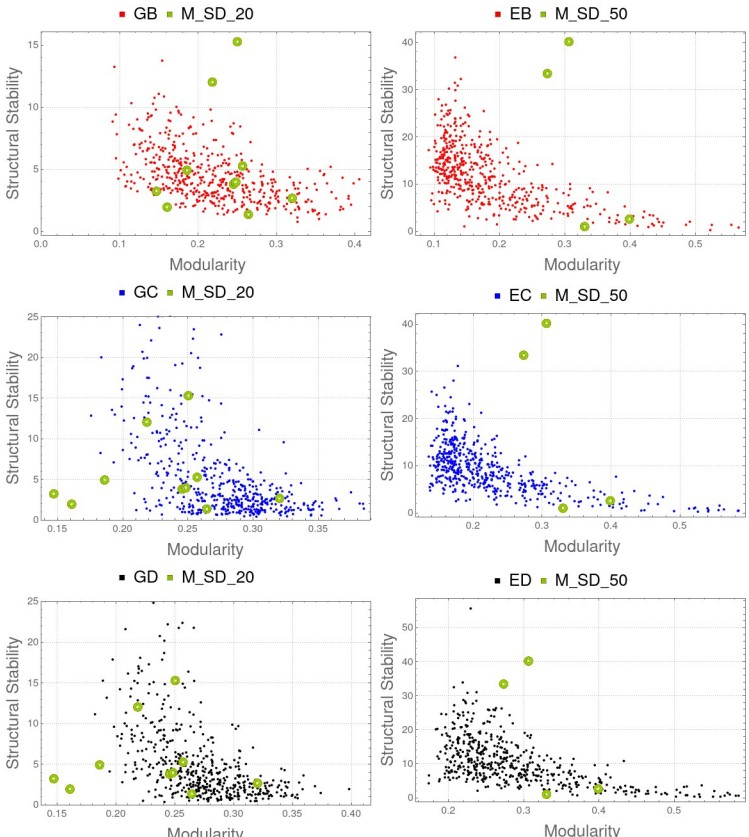

**Fig 11. Optimal modularity and structural stability.** Left (top to bottom): Comparison of `M_SD_20` with `GB`, `GC` and `GD`. Right (top to bottom): Comparison of `M_SD_50` with `EB`, `EC` and `ED`. Spearman correlation coefficient is lower than −0.057 in all these cases except `M_SD_20` (Table 2). It suggest that the formation of strongly interrelated communities within a biological system has a negative influence on maximal biological diversity.

way they are joined. All this information is usually belittled in many research contributions. ISs are described in our Lotka-Volterra system with precision, so allowing for further studies on robustness, bifurcation phenomena or metastability of solutions, in which the role of the associated informational field may be crucial [28, 33, 62, 72, 73].

The concept of structural stability is used in [22], and defined as in the present paper in [39, 74]. In this work we have introduced a global framework to study the structural stability for every possible stationary point of a mutualistic system. It is very important to determine the robustness of each asymptotic regime, measured by the region that associated parameters reach. In this sense, we have introduced an $\mathbf{R}^N$-partition for the $\alpha$-parameters describing the different convex regions for each stationary solution, which is moreover globally asymptotically stable in these regions. To our knowledge, this is the first time structural stability is used to study all the possible future scenarios, and not only to determine maximal biodiversity. We are aware we have not taken all the important information from the existence of an information structure. Indeed, there are many possible configurations possessing the same global asymptotic stable stationary solution (see for instance Fig 1 in which the set of semistable stationary points above the last asymptotically stable one could be very different), and would deserve further research.

The level of the interdependence between structure and dynamics on complex networks is not always clear; sometimes it seems that this relation induces determination, and usually just correlation in most cases. In this paper we conclude that dynamics, although closely related, is mainly determined by the signed sum of the degrees of the net and not for other parameters of the topology of the network.

To study the dependence of the topology and its associated dynamics we have introduced an $N$-dimensional Lotka-Volterra system of differential equations.

Our results also suggest that optimal modularity has a negative impact on biological diversity (structural stability), but in a smoother way that the positive influence of the sum of cooperation $\gamma_{ij}$ values. That is, the formation of strongly interrelated communities within a biological system has a negative influence on maximal biological diversity. More conclusive results on modularity may require more extensive studies, maybe normalizing the connectivity matrix of the networks to a constant centrality degree or constant sum of cooperation coefficients. That way, the influence of modularity could be better estimated. Similar studies have been done in [39, 43, 60], and they have shown that some network properties, such as nestedness, are also a secondary process (not a core process) shaping coexistence.

We conjecture that competition/antagonism $\beta_{ij}$ coefficients have negative influence on maximal biological diversity, but more extensive experiments could also be required to confirm this claim.

## Acknowledgments

Authors thank Prof. Francisco J. Esteban, at the Faculty of Biology at Jaen University (Spain) for their useful suggestions to improve a previous version of this paper. We also want to thank the Computational Center at the Computer Engineering High Technical School at Seville University and Jeśus Cano for technical asistence.

## Author Contributions

**Conceptualization:** José R. Portillo, Fernando Soler-Toscano, José A. Langa.

**Data curation:** José R. Portillo, Fernando Soler-Toscano, José A. Langa.

**Formal analysis:** José R. Portillo, Fernando Soler-Toscano, José A. Langa.

**Funding acquisition:** José A. Langa.

**Investigation:** José R. Portillo, Fernando Soler-Toscano.

**Methodology:** José R. Portillo, Fernando Soler-Toscano, José A. Langa.

**Project administration:** José A. Langa.

**Resources:** José A. Langa.

**Software:** José R. Portillo, Fernando Soler-Toscano.

**Supervision:** José R. Portillo, Fernando Soler-Toscano, José A. Langa.

**Validation:** José R. Portillo, Fernando Soler-Toscano, José A. Langa.

**Visualization:** José R. Portillo, Fernando Soler-Toscano.

**Writing – original draft:** José R. Portillo, Fernando Soler-Toscano, José A. Langa.

**Writing – review & editing:** José R. Portillo, Fernando Soler-Toscano, José A. Langa.

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
