## [Decision Letter · Decision Letter 0]

9 Feb 2022

PONE-D-21-36391Global structural stability and the role of cooperation in mutualistic systemsPLOS ONE

Dear Dr. Portillo,

Thank you for submitting your manuscript to PLOS ONE. After careful consideration, we feel that it has merit but does not fully meet PLOS ONE’s publication criteria as it currently stands. Therefore, we invite you to submit a revised version of the manuscript that addresses the points raised during the review process.

More precisely, you will see that the two reviewers are advising that you revise your manuscript. Note that it is a minor revision so consider making the suggested changes. 

We look forward to receiving your revised manuscript.

Kind regards,

Pablo Martin Rodriguez

Academic Editor

PLOS ONE

Journal Requirements:

3. We noted in your submission details that a portion of your manuscript may have been presented or published elsewhere. (This manuscript was previously submitted to a different PLOS journal as either a presubmission inquiry or a full submission.

PlOS Computational Biology

PCOMPBIOL-D-21-00722)

Reviewers' comments:

Reviewer's Responses to Questions

**Comments to the Author**

1. Is the manuscript technically sound, and do the data support the conclusions?

Reviewer #1: Yes

Reviewer #2: Partly

2. Has the statistical analysis been performed appropriately and rigorously? 

Reviewer #1: Yes

Reviewer #2: N/A

3. Have the authors made all data underlying the findings in their manuscript fully available?

Reviewer #1: Yes

Reviewer #2: Yes

4. Is the manuscript presented in an intelligible fashion and written in standard English?

Reviewer #1: Yes

Reviewer #2: Yes

5. Review Comments to the Author

Reviewer #1: In the current manuscript, Portillo et al. analyze the dynamics of ecological systems by using the concept of Global Structural Stability, which is introduced in the text. Structural stability is a well-established concept in dynamical system theory, and it refers to the robustness of a given system to have its dynamical behavior (number of equilibrium points, their stability, limit cycles, etc.) unchanged upon smooth variations in the parameters' configuration. In theoretical ecology, this notion is understood in a slightly different manner: a system is said to be more structurally stable than the other if it has a larger volume in the growth rates parameter space (defined by components $\\alpha_i$ of the growth rates vector $\\boldsymbol{\\alpha}$) in which the fixed point solution is feasible (maximal biodiversity) and stable. The Global structural stability introduced by authors is an extension of the latter concept in that it does not require the feasible point to be stable. In other words, global structural stability considers the volume in the growth rates parameter space that encompasses unstable and stable feasible points. The motivation for this definition stems from the fact that especially in high dimensional systems feasible solutions are seldom observed; and if they do occur, they appear along with semistable stationary points. Therefore, as it is argued in the manuscript, rich information about the dynamical system can end up being neglected by considering stable feasible points solely.

The idea of global structural stability is applied to plant-pollinator networks, which are mutualistic systems whose dynamics can be described by Lotka-Volterra models, as defined in Eq. (1). The abundance of the species depends on three factors: the intrinsic growth rate, competitive and mutualistic connections. The latter are encoded in a bipartite matrix, which, in the case of real networks, can be mapped by field observations. After presenting the idea of global structural stability, the authors investigate how that property is influenced by the strength of mutualistic interactions and structural characteristics of the networks.

The study put forward by the authors is interesting, the text is well-written, and the results appear to be solid. I believe the paper deserves publication in Plos One, but before that the authors should consider the issues I describe below.

The idea of Global structural stability is relevant to the study of ecological networks, but I found its presentation somewhat confusing. First, the motivation for the new concept is not clearly presented in the introduction part -- actually, one only starts to fully appreciate its importance and, more importantly, its difference from the standard structural stability after line 150, where global structural stability is discussed in more detail. In the introduction, the authors do mention that they "consider the whole set of stationary points (asymptotically stable, semistable, or even globally unstable), and not only the globally asymptotically stable point with all components positive (see [45, 47] for a similar approach)"; but, in my opinion, it is not evident to the reader that this is part of the new measure that is being introduced here. Therefore, I would suggest to highlight the relevance of the global structural stability, as it is done after line 160, right in the introduction section.

Second, Information Structure (IS) seems to be a crucial element in the analysis of the networks; however, in my view, this concept is not clearly defined in text. In the first paragraph of the manuscript, it is mentioned that "When the dynamics of the system is given by a set of differential equations, its behaviour generically depends on its global attractor [31–34], defined as information structure (IS) (...)". The latter sentence leaves the impression that IS is the very global attractor; if that is indeed the case, why does one need to relabel "global attractor" to IS? In Refs. [30,35] I notice that that the definition is more subtle; therefore, I would suggest to refine the definition of this concept.

In the Dicussion section, it is stated that:

"The level of the interdependence between structure and dynamics on

complex networks is not always clear; sometimes it seems that this relation induces determination, and usually just correlation in most cases. In this paper we conclude that dynamics, although closely related, is not determined by the topology of the network."

It is very difficult to categorically conclude that the network dynamics is not determined by its topology by analyzing the correlation of the structural stability with only three metrics (mean degree, nestdeness and modularity). There are other basic quantities, such as the number of nodes and the largest eigenvalue of the adjacency matrix, that could have a more clear or direct influence on the dynamics. Besides, in Fig. 9 we see a clear relation between mean degree and structural stability. So how can one conclude that the dynamics is not determined by the topology? Perhaps it would better to clarify what it is meant by "closely related" and "determined" in this context. Thus, I would suggest to either soften the statement in that paragraph or provide more analyses with other network measurements. Furthermore, concerning nestedness, there have been works showing that this index may not play the important role in shaping the network dynamics as it was previously believed [Phys. Rev. X 9, 031024 (2019)].

Minor points:

Line 309: "To build networks of the same size, we selected the plants and animals with more observed interactions." What do you the authors mean by "build the networks of the same size"? Are not the networks taken from the Web of Life already built when they are inserted into the dynamical equations?

Line 316: I believe the term "G* studies" has not been defined previously. One can figure it out by the context what is meant by the expression, but for the sake clarity I suggest the authors to explicitly define what class of experiments "G* studies" refers to.

Line 479: "Similar studies has" -> Similar studies *have*.

Between lines 480-482: It would be pertinent to cite Phys. Rev. X 9, 031024 (2019) along with Refs. [40,42], since the former shows that nestedness is actually an entropic consequence of the degree sequence of the mutualistic networks, and not an irreducibly macroscopic feature.

There are a few parts in the text with the connector "y", like in line 305: "23 y 121".

The caption of Fig. 1 could be improved: what is the meaning of the arrows between the network visualizations and the arrows connecting the nodes $u_i$ in the last two lines? The network in the upper left panel is probably the network over which the dynamics is performed, and its structure is undirected, as we can see in picture and in the adjacency matrix. However, it is not clear how the directed links on the right appear. Furthermore, if every small network on the right is a non-negative stationary point of the dynamics, how can the system transition to one state to the other as indicated by the arrows? If they are stationary, their configuration should remain stable in the long run. Probably this is related to the definition of IS, which, as previously mentioned, should be better defined in the text.

Caption of Fig. 5: The y-axis should indicate "Global structural stability" instead of "Structural stability" in order not to leave room for doubt about what is being plotted.

Caption of Fig. 9: In this figure, does the plot show the structural stability or the global one? If it is the former, then the labels in the axes and the caption match, otherwise they should be modified accordingly.

Caption of Fig. 10: Same comment as for the caption of Fig. 9.

Reviewer #2: Review PONE-D-21-36391

This manuscript presents a classic model of the Lotka-Volterra cooperative system. The authors investigate the dependence of a system's region of maximum biodiversity on its connectivity matrix, using a network of graphs. To study topology dependence and its associated dynamics, they introduced a system of N-dimensional Lotka-Volterra differential equations. They use the adjacency matrix associated with this system to describe the underlying structural graph. The Lotka-Volterra cooperative type systems assume a class in which, due to the positive input nature of the existing links between nodes, it maximizes the influence of the network topology on the dynamics.

I would like to suggest some minor revisions:

1. Figures 4, 5, 6, 10 and 11 should be adjusted to the format that all information can be read.

2. On lines 468, 469 and 470 the authors state that:

"Indeed, it is the class of systems from which the richness of the number of existing (cooperative) connections has a direct consequence on the plurality (diversity) of future scenarios." It is not clear where the authors prove this in the text.

3. The statements placed in parts "Discussion" are interesting, but no relationship is made with what was done in the text. Thus, the authors must refer where in the text the arguments that lead to the statements were presented.

6. PLOS authors have the option to publish the peer review history of their article (what does this mean?). If published, this will include your full peer review and any attached files.

Reviewer #1: No

Reviewer #2: No

---

## [Author Response · Author response to Decision Letter 0]

2 Mar 2022

Detailed answers to Academic Editors and Reviewers. 

Comments to the Authors

Reviewer #1: In the current manuscript, Portillo et al. analyze the dynamics of ecological systems by using the concept of Global Structural Stability, which is introduced in the text. Structural stability is a well-established concept in dynamical system theory, and it refers to the robustness of a given system to have its dynamical behavior (number of equilibrium points, their stability, limit cycles, etc.) unchanged upon smooth variations in the parameters' configuration. In theoretical ecology, this notion is understood in a slightly different manner: a system is said to be more structurally stable than the other if it has a larger volume in the growth rates parameter space (defined by components $\\alpha_i$ of the growth rates vector $\\boldsymbol{\\alpha}$) in which the fixed point solution is feasible (maximal biodiversity) and stable. The Global structural stability introduced by authors is an extension of the latter concept in that it does not require the feasible point to be stable. In other words, global structural stability considers the volume in the growth rates parameter space that encompasses unstable and stable feasible points. The motivation for this definition stems from the fact that especially in high dimensional systems feasible solutions are seldom observed; and if they do occur, they appear along with semistable stationary points. Therefore, as it is argued in the manuscript, rich information about the dynamical system can end up being neglected by considering stable feasible points solely.

The idea of global structural stability is applied to plant-pollinator networks, which are mutualistic systems whose dynamics can be described by Lotka-Volterra models, as defined in Eq. (1). The abundance of the species depends on three factors: the intrinsic growth rate, competitive and mutualistic connections. The latter are encoded in a bipartite matrix, which, in the case of real networks, can be mapped by field observations. After presenting the idea of global structural stability, the authors investigate how that property is influenced by the strength of mutualistic interactions and structural characteristics of the networks.

The study put forward by the authors is interesting, the text is well-written, and the results appear to be solid. I believe the paper deserves publication in Plos One, but before that the authors should consider the issues I describe below.

The idea of Global structural stability is relevant to the study of ecological networks, but I found its presentation somewhat confusing. First, the motivation for the new concept is not clearly presented in the introduction part -- actually, one only starts to fully appreciate its importance and, more importantly, its difference from the standard structural stability after line 150, where global structural stability is discussed in more detail. In the introduction, the authors do mention that they "consider the whole set of stationary points (asymptotically stable, semistable, or even globally unstable), and not only the globally asymptotically stable point with all components positive (see [45, 47] for a similar approach)"; but, in my opinion, it is not evident to the reader that this is part of the new measure that is being introduced here. Therefore, I would suggest to highlight the relevance of the global structural stability, as it is done after line 160, right in the introduction section.

We have moved the following paragraphs to the introduction: 

“The study of the size of the region for intrinsic growth parameters (in our case the α_i parameters) for which a system reaches its optimal biodiversity (all the species present) is defined as Structural Stability in [41]. This is a crucial fact for the study of the robustness of biodiversity in an ecosystem, as it characterizes the borders for intrinsic growth to get

maximal biodiversity. 

Following the Linear Complementary Theory (LCP) associated to

Lotka-Volterra systems [38, 39, 49], we introduce a partition of the phase space [50] for which we can estimate the area in which each stationary solution is globally stable, by measuring the intersection of its associated cone of biodiversity with the unit N-dimensional sphere [45]. 

However, specially in high dimensional systems, a stationary point with all its components strictly positive either does not exist, or, if this is the case, there also exists a big set of semistable stationary points. The presence of these stationary points is crucial for the description of the transient behaviour and metastability properties of the system, so that

neglecting its study could lead to wrong conclusions. Moreover, the ways to reach a particular stationary solution are multiple, depending of the different

(informational) landscapes [35] described in detail by its semistable solutions (see Figure 2).”

Second, Information Structure (IS) seems to be a crucial element in the analysis of the networks; however, in my view, this concept is not clearly defined in text. In the first paragraph of the manuscript, it is mentioned that "When the dynamics of the system is given by a set of differential equations, its behaviour generically depends on its global attractor [31–34], defined as information structure (IS) (...)". The latter sentence leaves the impression that IS is the very global attractor; if that is indeed the case, why does one need to relabel "global attractor" to IS? In Refs. [30,35] I notice that that the definition is more subtle; therefore, I would suggest to refine the definition of this concept.

We have refined the definition introducing the following paragraphs:

“Note that, for an autonomous system, an IS is just the detailed structure of the unique global attractor. In gradient systems, this IS induces a whole deformation of the phase space, drawing an informational landscape where the transient and asymptotic observed dynamics of the system hold [35] . This IS and informational landscape are fixed and attracting. As indicated above, they coincide with the global attractor. But, in non-autonomous systems, in which, for instance, parameters depend on time, this fixed structure and associated landscapes are also changing in time, loosing their invariance and attracting properties, but still being crucial for the description of the dynamics. This fact has been used, for instance, in Neuroscience to discriminate in detail subjects with disorders of consciousness [37]. Thus, and IS could not coincide with the standard definition of a global attractor as the object describing all the asymptotic behaviour of the system. This is why, even in an autonomous framework as we use in this paper, attractors and IS have to be differentiated.”

We have also added a new reference for clarifying the use of IS:

37. Galadi JA, Silva-Pereira S, Sanz Perl Y, Kringelbach ML, I G, Laufs H, et al. Capturing the non-stationarity of whole-brain dynamics underlying human brain states. Neuroimage. 2021;(244).

doi:10.1016/j.neuroimage.2021.118551.

 In the Discussion section, it is stated that:

"The level of the interdependence between structure and dynamics on

complex networks is not always clear; sometimes it seems that this relation induces determination, and usually just correlation in most cases. In this paper we conclude that dynamics, although closely related, is not determined by the topology of the network."

It is very difficult to categorically conclude that the network dynamics is not determined by its topology by analyzing the correlation of the structural stability with only three metrics (mean degree, nestdeness and modularity). There are other basic quantities, such as the number of nodes and the largest eigenvalue of the adjacency matrix, that could have a more clear or direct influence on the dynamics. Besides, in Fig. 9 we see a clear relation between mean degree and structural stability. So how can one conclude that the dynamics is not determined by the topology? Perhaps it would better to clarify what it is meant by "closely related" and "determined" in this context. Thus, I would suggest to either soften the statement in that paragraph or provide more analyses with other network measurements. 

Actually, our previous phrase is confusing. For some graph theorists, degrees are not part of the topology. Avoiding confusion, we changed the phrase by: "is mainly determined by the signed sum of the degrees of the net and not for others parameters of the topology of the network."

Furthermore, concerning nestedness, there have been works showing that this index may not play the important role in shaping the network dynamics as it was previously believed [Phys. Rev. X 9, 031024 (2019)]

We have included reference and added some words concerning nestedness role.

[60] Payrató-Borràs C, Hernández L, Moreno Y. Breaking the Spell of Nestedness: The Entropic Origin of Nestedness in Mutualistic Systems. Phys Rev X. 2019;9:031024.

doi:10.1103/PhysRevX.9.031024.

Minor points:

Line 309: "To build networks of the same size, we selected the plants and animals with more observed interactions." What do you the authors mean by "build the networks of the same size"? Are not the networks taken from the Web of Life already built when they are inserted into the dynamical equations?

We have added a phrase for clarifying the construction of the networks.

Line 316: I believe the term "G* studies" has not been defined previously. One can figure it out by the context what is meant by the expression, but for the sake clarity I suggest the authors to explicitly define what class of experiments "G* studies" refers to.

We have included two introduction phrases to clearing the mean. Thanks.

Line 479: "Similar studies has"  Similar studies *have*.

Done

Between lines 480-482: It would be pertinent to cite Phys. Rev. X 9, 031024 (2019) along with Refs. [40,42], since the former shows that nestedness is actually an entropic consequence of the degree sequence of the mutualistic networks, and not an irreducibly macroscopic feature.

Reference is included

There are a few parts in the text with the connector "y", like in line 305: "23 y 121".

Done. Thanks again.

The caption of Fig. 1 could be improved: what is the meaning of the arrows between the network visualizations and the arrows connecting the nodes $u_i$ in the last two lines? The network in the upper left panel is probably the network over which the dynamics is performed, and its structure is undirected, as we can see in picture and in the adjacency matrix. However, it is not clear how the directed links on the right appear. Furthermore, if every small network on the right is a non-negative stationary point of the dynamics, how can the system transition to one state to the other as indicated by the arrows? If they are stationary, their configuration should remain stable in the long run. Probably this is related to the definition of IS, which, as previously mentioned, should be better defined in the text. 

The caption of Fig. 1 has been revised

Caption of Fig. 5: The y-axis should indicate "Global structural stability" instead of "Structural stability" in order not to leave room for doubt about what is being plotted.

It is OK. Changed and thanks. Done 

Caption of Fig. 9: In this figure, does the plot show the structural stability or the global one? If it is the former, then the labels in the axes and the caption match, otherwise they should be modified accordingly.

It is the structural stability. It is OK

Caption of Fig. 10: Same comment as for the caption of Fig. 9.

It is the structural stability. It is OK

Reviewer #2: Review PONE-D-21-36391

This manuscript presents a classic model of the Lotka-Volterra cooperative system. The authors investigate the dependence of a system's region of maximum biodiversity on its connectivity matrix, using a network of graphs. To study topology dependence and its associated dynamics, they introduced a system of N-dimensional Lotka-Volterra differential equations. They use the adjacency matrix associated with this system to describe the underlying structural graph. The Lotka-Volterra cooperative type systems assume a class in which, due to the positive input nature of the existing links between nodes, it maximizes the influence of the network topology on the dynamics.

I would like to suggest some minor revisions:

1. Figures 4, 5, 6, 10 and 11 should be adjusted to the format that all information can be read.

We have improved all the figures and changed font sizes for clarity. For example, in Figs 9, 10 and 11, the real networks of sedd-dispersers animals vs. fruits are represented by the symbol of a fruit.

2. On lines 468, 469 and 470 the authors state that:

"Indeed, it is the class of systems from which the richness of the number of existing (cooperative) connections has a direct consequence on the plurality (diversity) of future scenarios." It is not clear where the authors prove this in the text. 

We have erased the unclear affirmation

3. The statements placed in parts "Discussion" are interesting, but no relationship is made with what was done in the text. Thus, the authors must refer where in the text the arguments that lead to the statements were presented.

We have simplyfied the text and we have erased the parts without a clear relationship.

Dr. José R. Portillo, on behalf of all authors

Sevilla, March 1st, 2022

PS:Previous version sended to J Comp Bio has been full revised and experiments has been improved with more networks and bigger size.

---

## [Decision Letter · Decision Letter 1]

8 Apr 2022

Global structural stability and the role of cooperation in mutualistic systems

PONE-D-21-36391R1

Dear Dr. Portillo,

We’re pleased to inform you that your manuscript has been judged scientifically suitable for publication and will be formally accepted for publication once it meets all outstanding technical requirements.

Kind regards,

Pablo Martin Rodriguez

Academic Editor

PLOS ONE

Additional Editor Comments (optional):

Reviewers' comments:

Reviewer's Responses to Questions

**Comments to the Author**

1. If the authors have adequately addressed your comments raised in a previous round of review and you feel that this manuscript is now acceptable for publication, you may indicate that here to bypass the “Comments to the Author” section, enter your conflict of interest statement in the “Confidential to Editor” section, and submit your "Accept" recommendation.

Reviewer #1: All comments have been addressed

Reviewer #2: All comments have been addressed

2. Is the manuscript technically sound, and do the data support the conclusions?

Reviewer #1: Yes

Reviewer #2: (No Response)

3. Has the statistical analysis been performed appropriately and rigorously? 

Reviewer #1: Yes

Reviewer #2: Yes

4. Have the authors made all data underlying the findings in their manuscript fully available?

Reviewer #1: Yes

Reviewer #2: Yes

5. Is the manuscript presented in an intelligible fashion and written in standard English?

Reviewer #1: Yes

Reviewer #2: Yes

6. Review Comments to the Author

Reviewer #1: I went over the correspondence, as well as the new version of the manuscript, and in my opinion the authors have successfully addressed the points in both reports. I therefore recommend the publication.

Reviewer #2: (No Response)

7. PLOS authors have the option to publish the peer review history of their article (what does this mean?). If published, this will include your full peer review and any attached files.

Reviewer #1: No

Reviewer #2: No

---

## [Editor Report · Acceptance letter]

11 Apr 2022

PONE-D-21-36391R1 

Global structural stability and the role of cooperation in mutualistic systems 

Dear Dr. Portillo:

I'm pleased to inform you that your manuscript has been deemed suitable for publication in PLOS ONE. Congratulations! Your manuscript is now with our production department. 

Kind regards, 

on behalf of

Professor Pablo Martin Rodriguez 

Academic Editor

PLOS ONE